# Using Partnerships and Community Science to Protect Wild and Scenic Rivers in the Eastern United States

**Alison Field-Juma [1],* and Nancy Roberts-Lawler [2]**

1 OARS: For the Assabet, Sudbury and Concord Rivers, Concord, MA 01742, USA
2 Musconetcong Watershed Association, Asbury, NJ 08802, USA; nancy@musconetcong.org
* Correspondence: afieldjuma@oars3rivers.org

**Abstract:** The Musconetcong (New Jersey) and the Sudbury-Assabet-Concord (Massachusetts) are federally-designated Partnership Wild and Scenic Rivers, a model for river conservation under the Wild and Scenic Rivers Act. These two rivers are embedded in a patchwork of private and public land ownership. The Act has been used to facilitate partnerships among municipal, state, federal and local non-profit actors to implement river conservation plans. These partnerships have supported community science-based monitoring to make the case for dam removal and stricter water pollution controls. Two case studies examine using community science to provide actionable data to decision-makers. In New Jersey, a documented increase in macroinvertebrates post-dam removal supported additional dam removals, leading to the return of American shad to the river. Quality controls and training proved to be key components. In Massachusetts, stricter effluent discharge permits reduced instream Total Phosphorus from 0.8 mg/L in 1999 to the eutrophication threshold of 0.023–0.05 mg/L. Community engagement in river science and stewardship was an important co-benefit. As many US rivers evolve from generating hydropower and conveying waste into major recreational resources, local organizations are uniquely positioned to engage the public and generate quality-controlled data to use in advocating for major improvements in water and habitat quality. Useful policy and regulatory frameworks for broader applicability are suggested.

**Keywords:** citizen science; Wild and Scenic Rivers Act; community science; dam removal; NPDES; water quality monitoring; Clean Water Act



## 1. Introduction

Rivers in the eastern United States have undergone a profound evolution in roles over the past three centuries. From initially providing abundant fish protein and transportation routes, colonial and early industrial era dams harnessed them for hydropower which severely limited the earlier uses. At the same time, rivers were used to convey waste from farms, homes, and factories to the ocean. The result was heavily polluted rivers with dams that blocked the migration of fish and accumulated contaminated and nutrient-rich sediments. This pattern has repeated, in various permutations, around the world [1].

Today, demands for cleaner water, changes in laws and policies to limit pollution, and major investments by municipalities in effluent controls, are transforming these rivers into valuable recreational and scenic resources. This progress would not have been possible without the evolution of the legal regime, in particular the passage of the federal Clean Water Act (CWA) of 1972 [2]. The CWA is implemented jointly by federal (national) and state environmental agencies: Environmental Protection Agency (EPA), and New Jersey Department of Environmental Protection (NJDEP) and Massachusetts Department of Environmental Protection (MassDEP), respectively. This shift away from conveying waste and generating hydropower required major public investment to reduce pollution and remove dams. In some cases, these investments have paid off handsomely in the return of migratory species like American shad (*Alosa sapidissima*) [3]. The Wild and Scenic Rivers Act (WSRA), and the resulting Partnership Wild and Scenic Rivers Program, is another

regulatory and institutional tool that aims to protect rivers from damage by dams and river pollution.

How can community science, coordinated by non-profit watershed-based organizations, help to develop durable protections for our free-flowing rivers in this changing legal and policy landscape? Does the involvement of many partners of different types at multiple levels create bureaucratic inefficiencies, or can it result in a better outcome? Palmer et al. identify improved monitoring of water and habitat quality as a top priority in order to provide the necessary data for management of valuable river assets [1]. Decisions that are based on credible data generated at the community level can be used to make decisions that are more likely to be effective than those with less scientific basis. By involving local stakeholders, community-generated data can also have important co-benefits including buy-in into the solutions and long-term resource stewardship. These studies will show that having accurate and defensible water quality data is crucial to selecting the best approach and technologies, securing major investment by local governments and the public, and achieving the desired effect, whether for pollution control or dam removal. Ultimately, when tackling point-source causes of water pollution the use of the right approach and technology to remediate a source can make the difference between success and failure. These cases also support the proposition that having multiple partners, including non-state actors and community members, can result in more effective river restoration and long-term management.

## 2. The Partnership Wild and Scenic Rivers Framework

### 2.1. Using the Wild and Scenic Rivers Act for Partnerships

Rivers in the northeastern and mid-Atlantic states were heavily dammed starting in the 1700s and continuing through the industrial revolution. The relatively small state of Massachusetts alone has more than 3000 dams [4]. In New Jersey, there are over 1700 dams [5]. Most are now obsolete, many are in disrepair and may pose a hazard, and many impair water quality. As a result, and with a renewed interest in migratory fish passage and recreation, these regions are in the midst of efforts to restore free-flowing conditions to rivers while protecting segments that remain unimpounded. A growing number of rivers in the eastern U.S. are being designated as "Wild and Scenic", despite the population density of the region and relative lack of wilderness. This designation is a unique tool that brings attention to the need for durable river protection. It has given non-profit organizations an opening to have an effective role in science and advocacy.

The U.S. Congress enacted the National Wild and Scenic Rivers Act in 1968 to protect and enhance the outstanding natural, cultural and recreational values of certain free-flowing rivers into the future [6]. Jennings notes that the act "is nearly unique in requiring the improvement of a protected natural resource's integrity, function, or condition" [7] (p. 21). The Wild and Scenic Rivers System was "a visionary template for a collaborative approach to river protection involving federal, state, and local partners and is the nation's primary river conservation authority" [7] (p. 16). Most of the 226 designated Wild and Scenic rivers, totaling over 20,921 km, are located in large tracts of public lands under the control of a single federal agency (U.S. Forest Service, U.S. Fish and Wildlife Service, Bureau of Land Management, or National Park Service). For some, river management is the responsibility of states or tribes [8]. However, the patchwork of smaller public and private land holdings in the eastern United States required a different approach.

The Partnership Wild and Scenic Rivers Program, administered by the Secretary of the Interior through the National Park Service, was developed to reflect this management challenge [9]. The program formalizes a partnership of local, state and federal government representatives and non-profit watershed partners, to implement a River Conservation Plan approved by Congress for each river. Federal funding is provided annually for this implementation that has been devolved to the local level. In 2020, Congress allocated $3.5 million to be equally divided among the 17 partnership rivers, an increase over time from $1.7 million allocated in 2011 for 12 partnership rivers [10]. In 2007, the Ash

Institute for Democratic Governance and Innovation at Harvard Kennedy School named the Partnership program one of the top 50 government innovations linking citizens with important public services. Only six federal programs were selected for the award [11].

What makes this approach particularly durable, is that the designation of each river segment requires the formal approval of the government of every affected municipality. This is a rigorous process: a detailed "designation study," stakeholder engagement, and approval by the Town Meeting or City Council of each municipality. To gain approval, the study has to select river qualities that the local population values and, in some cases, convince them that this would not be a federal government "takeover" of their land. In 1999, 46.7 km of the Sudbury, Assabet and Concord Rivers were designated Wild and Scenic for their "outstanding ecology, history, scenery, recreation values, and place in American literature" [12]. In 2006, 39 km of the Musconetcong River were designated for recreational fishing, scenic views, wildlife habitat, and unique history [13]. There are now 17 rivers in the Partnership Wild and Scenic Rivers program, from Florida to Vermont.

The Act requires that the river is "free-flowing". However, the existence of small dams does not immediately disqualify a river segment from being added to the national Wild and Scenic Rivers system. While impounded river segments cannot be classified as "wild" or "scenic", they can be classified as "recreational" (Section 1273(b)(3)). This is the case for the Musconetcong which contains old "run-of-river" dams with little or no storage that do not significantly alter the flow regime.

## 2.2. Using the Clean Water Act to Protect Wild and Scenic Rivers

The Clean Water Act is one of the most important tools available for river protection, including designated Wild and Scenic rivers, in the United States. Yet by 1995, there were still insufficient data available to determine whether CWA goals were being achieved nationally, whether increasingly stringent point source effluent controls were yielding the expected benefits, or the role of non-point sources of pollution (such as stormwater) [14]. In the 1990s, water quality monitoring required under the Act was expanded to add ambient water quality and trends to the existing point-source discharge monitoring for compliance purposes. Monitoring of ecological conditions to assess aquatic system health still lagged [14]. Additional resources were needed to produce the required data. At the same time, partnerships began forming between federal and state agencies to work on water quality.

In 1994, the Massachusetts Watershed Initiative (MWI) was launched. This successful initiative was a broad partnership of state and federal agencies, watershed associations, businesses, municipal officials and individuals who worked together to improve the quality of the river corridors, drinking water, wetlands, wildlife, and other natural resources in the Commonwealth [15]. However, the 2008 recession introduced over a decade of relentless state budget cuts that significantly curtailed the robust watershed-based state water quality assessments that had emerged. In response the MassDEP began to provide technical support to watershed/river nonprofits to develop and manage water quality monitoring programs as a more cost-effective approach. In New Jersey in 2003, federal and state agencies formed the New Jersey State Water Quality Monitoring Council to limit monitoring redundancy, exchange data, and create partnerships [16]. As data exchange needs grew, the Council recognized that data produced for specific projects by universities and non-profits needed to be included in data exchange efforts.

## 2.3. The Niche for "Community Science" in Water Quality Monitoring

There is a long history of local watchdog groups monitoring water quality for the purpose of river protection, including collecting biological data. The Izaak Walton League, founded in 1922 by a group of anglers, was a pioneer in this arena and has the longest history of promoting water quality monitoring by volunteers through its Save Our Streams program [17]. The League sees itself at the forefront of efforts to limit water pollution and was one of the groups providing the vision behind the 1968 Wild and Scenic Rivers Act [18].

By the 1990s, the role of "citizens" in volunteer water quality monitoring throughout the United States was commonly recognized [19]. The term "citizen science" has been used in other scientific fields to describe science accomplished with public participation [20]. Although now in common usage, this term still does not adequately describe many of these high-quality monitoring programs that support advocacy for policy and river protection. "Community science" has been offered as encompassing more community-driven priorities rather than just collaboration with scientists [21]. In the current paper, the authors use the term "community science" because it is more inclusive, particularly where volunteers may not have the legal status of citizenship. This terminology is being adopted increasingly by U.S.-based environmental organizations to avoid the limitation and exclusiveness of the word "citizen" [22]. The authors recognize that this distinction may not be applicable in all regions of the world.

Loperfido et al. note that volunteer water quality monitoring was successful in "identifying and classifying most of the waters which violate United States Environmental Protection Agency recommended water quality criteria for total nitrogen (66%) and for total phosphorus (52%)" and was an important screening tool [23]. Once error and bias had been accounted for, these data could be used in Total Daily Maximum Load (TMDL) pollution budgeting or state water quality reporting (the authors found higher false negative (33–47%) and very low false positive reporting (1%)). In New Jersey, with 10,380 river km to cover, state biologists trained community members to collect samples for analysis by state-certified laboratories [24]. After the projects ended, some watershed associations continued monitoring with local funding. State funding through the MWI supported technical assistance for water quality assessments and, for the Assabet River, developing quality assurance, starting the TMDL process, initial sampling, and training of "citizen scientists" [25]. After the Initiative was eliminated in 2003, some watershed nonprofits were able to step in and continue water quality monitoring.

Over time it became clear to state agencies that data collected and managed by watershed associations were often more fine-grained and comprehensive than what the state could collect given their budget constraints. The barrier for regulatory use of these data was often data quality. Nationally, while watershed associations may have collected the data properly, they did not always analyze study results, update their study design, or share the data adequately. Ward made the case in 1996 that the taxpayer deserved water quality data collected using standardized definitions, data analysis and reporting methods, that was properly shared with the public [26]. In some cases, lack of quality control has meant that data were not used for regulatory decision-making. The EPA also required any project receiving federal funding for monitoring to have a Quality Assurance Project Plan (QAPP). By offering training, encouraging the development of QAPPs to define the methods and level of rigor, and supporting efforts to improve data sharing, the EPA and state environmental agencies found they could have more data resources to address water quality impairments. In both Massachusetts and New Jersey, the modification of data management systems to allow efficient data sharing between the EPA and state databases and to accept outside data is now operational.

In both states, the agencies encouraged those watershed associations that had volunteer water quality monitoring programs and the capacity to develop QAPPs [27]. In the case of New Jersey, NJDEP offered quality assurance training, and ruled that if NJDEP approved the QAPP, the agency would include the data in the regulatory review process. Watershed associations in both states began to use data they generated to support advocacy for river protection and river restoration. In 2019, EPA Region 1 offered quality assurance training for organizations running community science programs [28].

## 3. Establishing Rigor and Trust in Community-Based Data Collection

Two case studies of local watershed organizations in New Jersey (NJ) and Massachusetts (Mass.) are used to compare the uses of community science, as shown in Table 1. Both watersheds contain federally-designated Partnership Wild and Scenic (W&S) Rivers.

**Table 1.** Characteristics of the case study watersheds and watershed organizations compared.

| Characteristic | Musconetcong, New Jersey | SuAsCo, Massachusetts |
| --- | --- | --- |
| Watershed area/river length | 404 sq km/74 km | 1036 sq km/142 km |
| Year designated W&S | 2006 | 1999 |
| Outstanding W&S Values | Recreational fishing, scenic views, wildlife habitat, and unique history | Ecology, scenery, recreation, history, place in American literature |
| Km designated W&S | 39 + 6.9 (pending) | 46.7 |
| River Council partners | 14 + 1 (pending) municipal, 4 counties, 1 watershed non-profit, 2 other non-profits | 2 federal, 1 state, 8 municipal, 1 watershed non-profit, 1 land trust non-profit |
| Year watershed association founded | 1992 | 1986 |
| Non-profit staff/science volunteers (2019–2020) | 7 staff/25 volunteers 2 part-time science staff | 4 staff/41 volunteers 1 part-time science staff |
| Year QAPP developed | 2010 | 2000 |
| Data types, parameters | Percent EPT [1], macroinvertebrates, photographs of streambed and banks, habitat | Temp., pH, dissolved oxygen, conductivity, total suspended solids, total phosphorus, orthophosphate, nitrate-N, ammonia-N, chloride, chlorophyll-a, water depth, plant biomass |
| Monitoring funding sources | National Park Service (NPS), The Watershed Institute, member contributions | NPS, State legislature/MassDEP, Cedar Tree Foundation, environmental penalty, member contributions |

[1] Percent Ephemeroptera, Plecoptera and Trichoptera, adjusted for Hydropsychidae (% EPT) is a New Jersey HGMI metric, a gauge of sensitive species present.

*3.1. Case Study 1: Monitoring Water Quality Impacts of Dam Removal*

3.1.1. The Musconetcong River Setting

The Musconetcong River in northwestern New Jersey is the second largest tributary of the Delaware River. The river is one of the best trout streams in the state and flows through a valley with remarkable rural views, historic districts, and archeological sites [29]. In 1997, 18 of the 19 municipalities that border the river requested that the National Park Service determine the river's eligibility for Wild and Scenic designation [30]. In 2006, two segments, 39 eligible km, were designated as a Partnership Wild and Scenic River, one of only two in the state. The remaining eligible 6.9 km were excluded because one municipality had not voted for inclusion. The Partnership—the Musconetcong River Management Council—comprises federal, state and local municipal representatives as well as three non-profit organizations, including the Musconetcong Watershed Association (MWA). The MWA coordinates the Council, which works to implement the Congressionally-approved Musconetcong River Management Plan.

The Musconetcong River has a 13,000-year recorded history of human use supported in part by migratory fish such as American shad (*Alosa sapidissima*). In the 1700s, European settlers dammed the river and built water-powered mills. Blocked from their natural habitat, the shad stopped spawning in the Musconetcong. In addition to blocking movement of riverine species, some dams fundamentally altered the natural flow regime, water temperature and chemistry, sediment transport, and floodplain vegetation communities [1]. In the early 20th century, mills adopted electric power and the dams became obsolete, no longer maintained, and many were breached by floods.

By the early 2000s, of the over 1700 dams in New Jersey, 563 dams were rated high or significant hazard and many were impacting the river ecology [31]. Since only one dam had ever been removed for the purpose of restoring river flow in the state, NJDEP had no removal process to follow, nor recommended approach to evaluate restoration of aquatic health. Dam owners faced liability for dam failure and/or prohibitive permit and construction costs for repair.

The Musconetcong River Management Plan discusses the detrimental effects of dams on riverine character [32]. Identifying dams for removal was a Key Action in the Plan (Objective 1.1): "to protect and enhance the fisheries of the river and river tributaries." One run-of-river dam in the lower Musconetcong that was rated hazardous, the Finesville Dam, was in the excluded section of the Musconetcong River eligible under WSRA. Two of the high hazard dams, the Warren Glen and Hughesville Dams were upstream of that section, and affected water quality and fish migration. Repair or rebuilding of dams was not in keeping with the Plan.

In 2006, local, state and federal stakeholders explored river restoration options and a Musconetcong River Restoration Partnership (MRRP) was formed. By 2009, MWA had coordinated removal of remnants of two dams, only the second and third dams to be removed in the state. The Partnership applied for federal grants to remove the four larger dams in the lower Musconetcong that impeded fish migration. The Finesville, Hughesville, and Warren Glen removals would be coordinated by MWA, and removal of a dam upstream of Warren Glen, the Bloomsbury dam, would be coordinated by the U.S. Army Corps of Engineers. These grants required a public approval process, including meetings with local stakeholders. Public comments demonstrated concern that dam removals not harm the river or the community. Funders also required measures of success to justify the expenditure of multiple millions of dollars of public funds.

### 3.1.2. Using Data to Build Community Support

The MWA was founded in 1992 by local residents concerned about how the watershed's resources were being managed by state and local governments. The founders understood that durable protections would take many years and require a knowledgeable, engaged community supporting local, state and federal decision-making. MWA developed a river protection strategy that involved community members in collecting water quality data, and provided for educational seminars for property owners, teachers and local officials [33]. In 2009, MWA hired staff to develop a water quality monitoring process for dam removals. The goal was to answer partner and public questions about aquatic health and biological condition. Biological condition in streams is reflected by the presence of benthic macroinvertebrates [34]. Monitoring benthic macroinvertebrates before and after the dam removal was the most cost-effective way to answer questions and provide project assessments to funders.

### 3.1.3. Approach

The Finesville Dam removal presented MWA with an opportunity to develop a monitoring process as a pilot project. The public were concerned that the removal would negatively impact historical values and adjacent property owners were concerned that it would also damage river health and local fishing [35]. In response, MWA developed a plan to protect the historic values and a pre-removal water quality monitoring study.

MWA staff had trained volunteer monitors to sample macroinvertebrates to be identified to the Order or Family taxonomic level. However, genus or species-level data would be more useful since some aquatic insect larvae have specific feeding habits that might be limited by habitat availability [36]. Knowing whether these species were present would be useful as the practice of restoration monitoring evolved. In addition, NJDEP indicated that if the MWA developed a Quality Assurance Project Plan, and the state approved it, the state could use the data for regulatory purposes and water quality assessment. This would require MWA to send collected macroinvertebrate samples to a certified taxonomic laboratory for analysis. The laboratory would compile a report using the state index (NJ HGMI) to determine biological condition of the sites.

Since MWA's goal was to share the data with partners and other scientists, possibly engage the community, influence NJDEP decision-makers and use the data to assess project success to guide future work, MWA developed a QAPP, which was approved by NJDEP in 2010. The QAPP used the NJ HGMI score [37], Biological Condition assessment (Excellent,

Good, Fair, Poor), and percent EPT, to assess both water quality indicators and biological condition of the site. The QAPP described training, data review, and how data and scores would give information about the current condition of the sites and whether the dam removal had negatively impacted aquatic habitat and water quality.

Assessing the impoundment presented a challenge. Standard protocols for assessing biological conditions in streams are designed for wadeable areas, but the Finesville impoundment was typically 1.5 m deep. The study had to acknowledge that the habitat in the impounded area (Site 2), currently more like a lake, was likely to drastically change and return to more riverine conditions post-removal. Approximately 2 sq m of the most productive habitat would be sampled at each site. In wadeable sites, the streambed would be sampled using a kick method. For those too deep to wade, a dredge would be used to sample the stream bed from a boat. Sample analysis would be the same at all sites.

Sites were selected to represent existing conditions and habitat (see Figure 1):

Site 1: Wadeable local reference site to represent more natural riffle-run conditions upstream of planned in-stream work.

Site 2: Non-wadeable site within the impoundment with run, no riffle habitat, likely to be impacted by the dam removal.

Site 3: Wadeable site with riffle-run habitat, downstream of the dam removal.

Site 4: Wadeable site with riffle-run conditions downstream of the dam removal, not likely to be affected by in-stream work.

In April 2010, MWA volunteers, trained in the sampling protocols and habitat assessment using the EPA Visual Habitat Survey [38], collected and preserved pre-removal composite samples at all sites. Volunteers collected habitat information including underwater photographs in the impounded areas. Macroinvertebrate species in samples were identified, the NJ HGMI score and submetrics (including percent EPT) were calculated, and Biological Condition assessed (Table 2). Percent EPT is an indicator of sensitive species that support trout [39]. Photographs of the land, bank, and streambed (underwater) were taken before and after the dam removal.

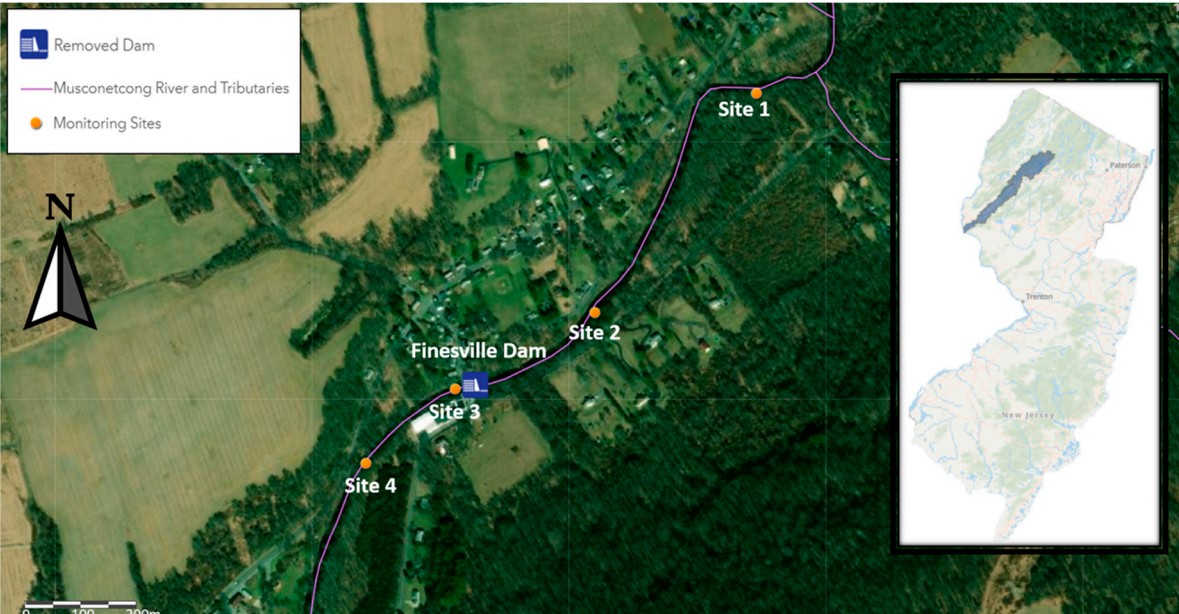

**Figure 1.** Finesville dam removal project area, showing monitoring sites relative to the dam location (40.6054° N, 75.1710° W) and locus of watershed in state (inset). Site 1 is the most upstream location.

**Table 2.** Finesville dam removal project monitoring results, showing conditions pre-dam removal (2010) and post-dam removal (2012).

| Assessment | Site 1 | Site 2 | Site 3 | Site 4 |
|:---:|:---:|:---:|:---:|:---:|
| **Pre-removal, 2010 Spring** | | | | |
| Biological Condition | Excellent | Poor | Excellent | Good |
| NJ HGMI | 67.25 | 17.53 | 69.64 | 56.41 |
| % EPT | 76.59 | 32.54 | 53.97 | 62.12 |
| Habitat Score | 159 | 32 | 110 | 135 |
| Habitat Assessment | Sub-optimal | Poor | Sub-optimal | Sub-optimal |
| **Post-removal, 2012** | | | | |
| Biological Condition | Excellent | Excellent | Excellent | Good |
| NJ HGMI | 78.69 | 80.61 | 72.93 | 57.02 |
| % EPT | 70.23 | 66.30 | 54.61 | 45.02 |
| Habitat Score | 158 | 77 | 100 | 125 |
| Habitat Assessment | Sub-optimal | Marginal | Sub-optimal | Sub-optimal |

Source: Unpublished MWA data, 2012.

The Spring 2010 data showed that biological condition pre-removal in the impoundment (Site 2), was Poor, and the NJ HGMI score and percent EPT were lower than at the other sites. The Habitat score and Assessment also indicated poor quality compared to other sites. Underwater photographs revealed that the rocky streambed was covered with sediment, poor habitat for aquatic life. MWA reported that, since the biological condition of the impounded areas was poor, dam removal was unlikely to cause increased harm to aquatic life in the impounded segment.

Local concerns abated after MWA data showed that fish habitat was already poor and might improve over time. The dam was finally removed in November 2011. In April 2012, volunteer collected post-removal data that showed that biological condition in Site 2, now wadeable, had changed to Excellent, percent EPT had improved and the Habitat scores improved from Poor to Marginal. Underwater photographs showed that rocky streambed was visible at Site 2, indicating that sediment had migrated downstream. Scores and assessments at Sites 1 and 3 were more similar to pre-removal conditions. Scores declined at Site 4, the most downstream site; underwater photographs and habitat assessments indicated an increase in sediment at the site.

An unexpected outcome was that biological condition in the impounded area had improved after only five months [40]. Restoration practitioners and engineers had previously asserted that biological condition would probably not improve for many years. MWA shared the data with the MRRP partners, who used this information to advocate for the next three dam removals, showing that well-planned and well-engineered dam removals did not negatively impact the food chain that supported sport fish like trout.

The data suggested that sensitive species absent from impounded areas might more easily colonize post-removal than expected and that American shad, alewife (*Alosa pseudoharengus*), or blueback herring (*Alosa aestivalis*) might return to the Musconetcong in the next several years. Within a few months, local fishermen reported that striped bass (*Morone saxatilis*), a fish known to follow American shad runs, were caught at the base of the Hughesville dam, the next dam upstream, approximately 3.8 km above the Finesville dam site.

*3.2. Case Study 2: Monitoring Water Quality Impacts of Wastewater Treatment*

3.2.1. The Sudbury, Assabet and Concord Rivers Setting

The Sudbury, Assabet and Concord Rivers (SuAsCo) form a single 1036 sq km watershed in central Massachusetts that drains northward into the Merrimack River in the

city of Lowell. The Merrimack flows eastward into the Atlantic Ocean at Newburyport, entering the Gulf of Maine. In 1999, 46.7 free-flowing km of these three rivers were federally designated as a Partnership Wild and Scenic River, the second in the state to receive this designation (Figure 2). The partnership comprises representatives of federal, state and local government (eight municipalities), plus the local watershed organization (OARS) and land trust, all working together as a River Stewardship Council. The Council is charged with implementing the River Conservation Plan, approved in the designation process by the eight municipalities and Congress [41].

The rivers were designated for five "outstandingly remarkable values": ecology, scenery, recreation, history (the start of the American Revolution), and place in American literature (Thoreau, Hawthorne, and Emerson). Yet at the time of designation the water quality was so poor in most segments of the Assabet and Concord rivers that they were on the State's List of Impaired Waters (303(d)) under the Clean Water Act, failing to meet the Class B water quality standard for nutrients, with eutrophication severely degrading both scenery and recreation [42].

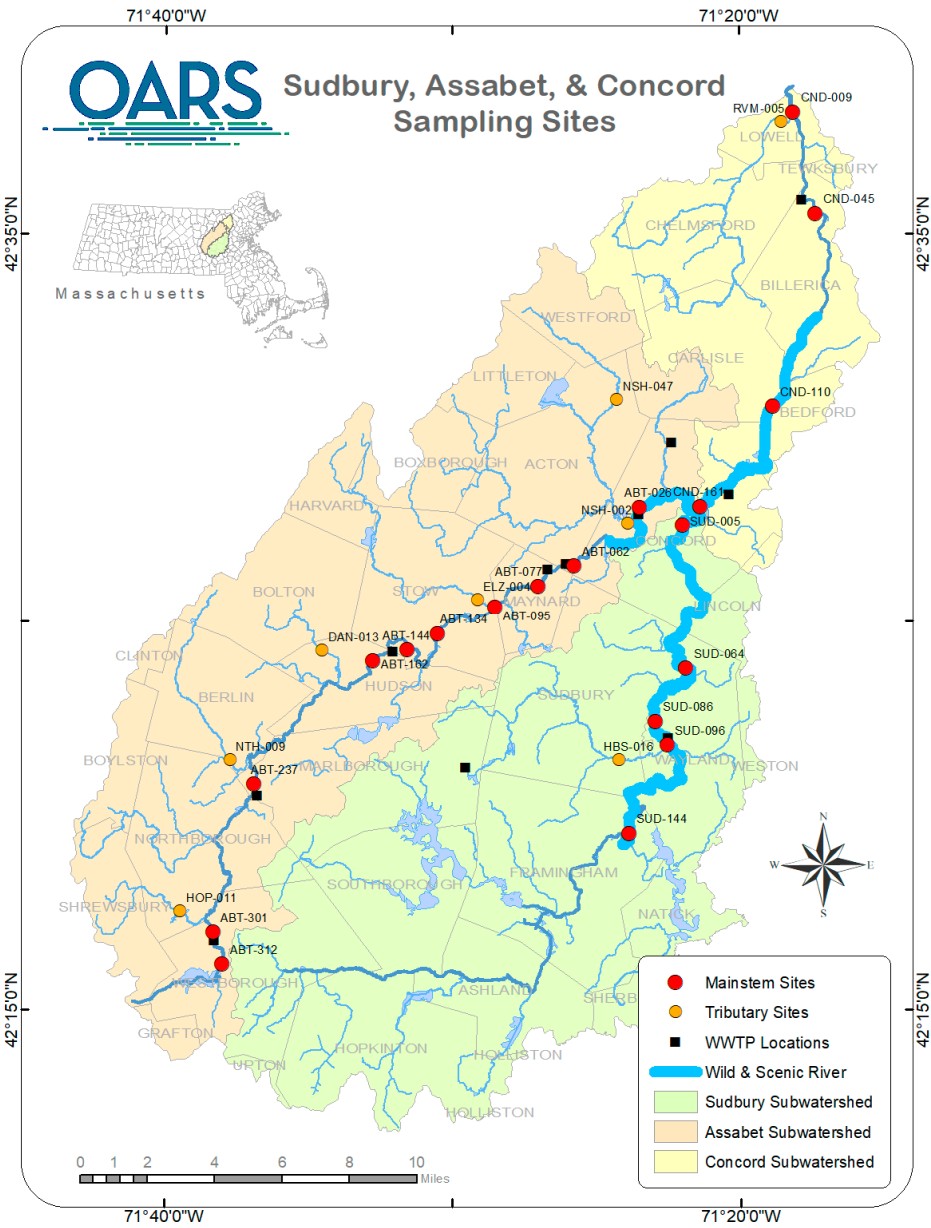

**Figure 2.** SuAsCo watershed showing wastewater treatment plants, water quality sampling sites, and designated Wild & Scenic River.

### 3.2.2. Adding Science from the Ground Up

The Organization for the Assabet River (OAR) was founded in 1986 by anglers, boaters, hunters and conservationists in order to clean up what was dubbed "The Cesspool of Massachusetts." OAR later added the Sudbury and Concord Rivers and adopted its current name, OARS. OARS started an informal volunteer-run water quality monitoring program in 1992 with state funds to help cover lab fees. The goal was to establish baseline river conditions and then assess the impact of wastewater treatment plant upgrades to reduce nutrients in their effluent. Beginning in the late 1980s, the monitoring raised awareness that excessive phosphorus was causing the Assabet's nutrient problem. This pointed to the need for stricter limits on the amount of phosphorus allowed under the discharge permits for four municipal wastewater treatment plants. Possible removal of the Assabet River from the Impaired Waters List, without proof that water quality standards had been met, showed clearly the need for quality-assured data documenting the existing impairments. Having documentation of in-stream phosphorus concentrations and eutrophication (biomass) levels would enable OARS to advocate for a Total Maximum Daily Load (TMDL) pollution budget study for phosphorus, the controlling nutrient in the river.

By 1999, OARS had written a QAPP, subsequently approved by the EPA and MassDEP. Importantly, data collected under the approved QAPP could be used in making regulatory decisions. OARS had a significant role in the MassDEP-sponsored TMDL study of the Assabet in 1999 [43]. OARS then set up a community science program that would annually train and run a team of up to 35 water quality volunteers. Their EPA-funded Assabet River StreamWatch project was a project of a partnership of municipalities, OARS, the US Geological Survey (USGS), the Massachusetts Division of Fisheries and Wildlife, and the Massachusetts Audubon Society [44].

### 3.2.3. Approach

By 2002, OARS had expanded water quality monitoring to 12 mainstem and nine tributary sites. Mainstem sampling sites were selected with one upstream and one downstream of each wastewater treatment plant. Monthly samples are collected in March, June-September, and November every year. The parameters are shown in Table 1 (under SuAsCo). Monitoring of *E. coli* bacteria was added subsequently [45] (p.8).

The TMDL for phosphorus (2004) concluded that "the Assabet River receives an excess of the nutrients phosphorus and nitrogen resulting in nutrient saturation and excessive growth of aquatic vegetation" and that point sources were the dominant source of biologically-available phosphorus [46]. It required an initial reduction of total phosphorus in treatment plant effluent from 0.75 mg/L to 0.1 mg/L from April through October, and a 90% reduction of phosphorus flux from sediment. By providing the necessary data, OARS contributed to the development of new, stricter, discharge permits issued by the EPA and MassDEP to the four wastewater treatment plants under the federal and state Clean Water Acts in 2005. When three municipalities appealed the stringent phosphorus concentration limits OARS also appealed, stating that the limits were insufficient to ensure attainment of Water Quality Standards. An agreement was reached to allow the new permits to go into effect with the stringent limits, with a joint letter from the EPA and MassDEP to the municipalities alerting them to prepare for even lower limits if needed and invest in "scalable technology."

The low Total Phosphorus (TP) concentration limit of 0.1 mg/L during the growing season set a new precedent for the six New England states in EPA's Region 1. Prior to these limits, the lowest municipal wastewater treatment plant discharge limits in New England were 0.2 mg/L TP. The municipalities invested significantly to add new phosphorus removal technologies to their treatment plants, with all upgrades completed by 2012 Each plant used a different P-removal technology—Comag, AquaDAF, BluPro, and Actiflow—based on their waste stream characteristics, cost and other factors. As a result, an 80% reduction in annual median concentrations of TP in the effluent was achieved, with annual median loads reduced by up to 91% [47]. In reference to the Hudson plant, the EPA noted:

"The upgrades have also improved the consistency of the treatment, enabled the WWTF to more easily meet its copper effluent limits, and reduced effluent concentrations of total suspended solids (TSS), biological oxygen demand (BOD), ammonia, and copper" [44] (p. 4). Similar co-benefits were experienced at the other three plants.

Guided by the TMDL, the regulators took an "adaptive management" approach to meeting water quality standards. Due to the complexity of the river system, including a legacy of nutrients deposited in sediment behind several dams, the permitting was divided into two phases. Phase 1 limits reduced growing season phosphorus limits to 0.1 mg/L, and a dam removal study was commissioned to determine how to manage phosphorus flux from the sediment. Phase 2 limits were to be based on the results of water quality and biomass monitoring, as per the TMDL. By 2012, OARS showed that the TP concentration in the water column in the free-flowing sections was approaching the target eutrophication threshold of 0.023–0.05 mg/L (Figure 3).

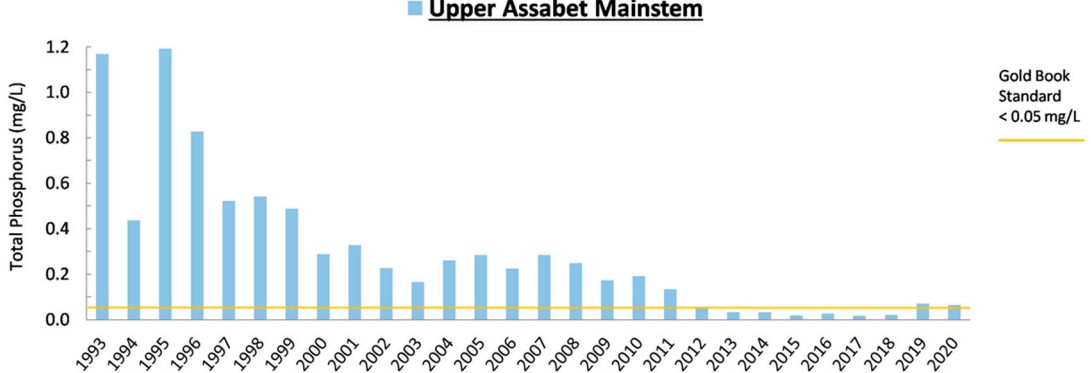

**Figure 3.** Mean summer Total Phosphorus Concentrations in Upper Assabet River declined 1993–2019 [45] (p. 39).

MassDEP monitored duckweed (*Lemna minor* and *Wolffia* spp.) a tiny, simple floating plant, as a proxy for the biomass in the impounded sections of the river [48]. In contrast, OARS monitored all biomass, including filamentous green algae and rooted aquatic plants that drew their nutrients from the sediment. Although the state's data showed a decrease in duckweed biomass, OARS' data showed a transition from duckweed to filamentous green algae, and an overall increase in biomass. The original problem—too much biomass in the river—was not getting better (Figure 4). Nutrients deposited in the deep sediments were still fueling eutrophication, possibly exacerbated by other factors like drought and high summer temperatures. MassDEP recognized that OARS' data demonstrated that the problem had not yet been solved.

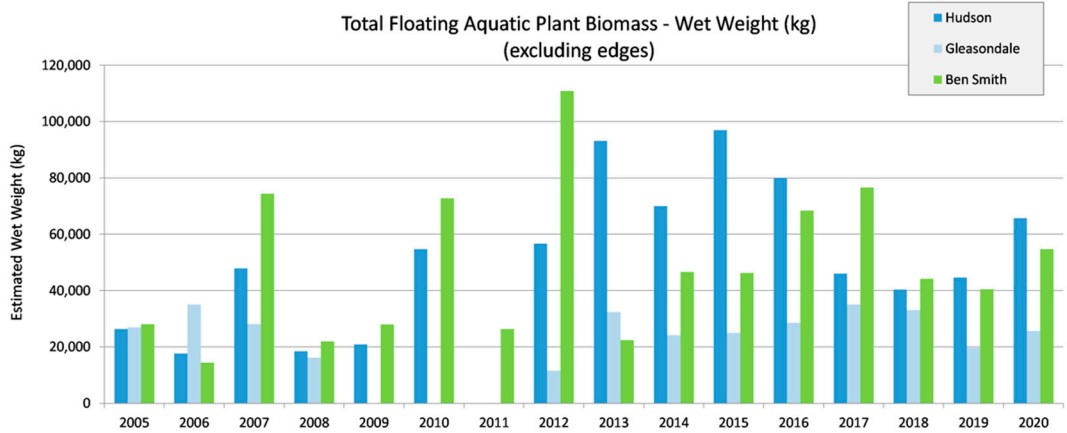

**Figure 4.** Total floating biomass in Assabet River impoundments increased, 2005–2019. Source: OARS Biomass Assessments (excluding emergent vegetation). Source: OARS, unpublished data.

The role of non-point sources of phosphorus could not be ignored. The TMDL required a 90% reduction in phosphorus flux from the sediment through dam or sediment removal, which had not been undertaken [46]. The state and EPA were working to reduce nonpoint sources of nutrients from stormwater but the problem of excessive biomass remained. Importantly, the dam removal study's modeling had shown that, contrary to earlier assumptions, the sediments did adsorb phosphorus during the non-growing season and release it during the growing season [49]. In 2019 and 2020, new (Phase 2) NPDES permit drafts were issued by the EPA that further reduced TP discharges in the non-growing season (November-March) from 1.0 mg/L to 0.2 mg/L to address this problem.

### 3.2.4. The Partnerships That Bring Progress

None of this would have been possible without partnerships and strong science-based advocacy. The partners included the federal EPA, National Park Service, and US Fish and Wildlife Service (USFWS), MassDEP, the Wild and Scenic River Stewardship Council, all the municipalities along the rivers (Assabet Consortium, public works departments, conservation commissions, chief executives, treatment plant operators), the Conservation Law Foundation and other non-profits, state and federal legislators, and the community science volunteers. Occasionally advocacy pitted partners against each other—such as when OARS successfully appealed the three permits in 2004 and a municipal permit change in 2008. The resulting precedent was then used to successfully argue a citizen suit against a NPDES wastewater permit for a discharge directly on the Sudbury Wild and Scenic River segment. Eventually, all parties worked together to address the shared goal of clean water, recognizing the need to base decisions on good data and the necessity of trade-offs (e.g., higher sewer fees).

Over the past decade, MassDEP's budget for water quality monitoring and TMDLs has repeatedly been cut. The ongoing need for water quality monitoring, however, provided a niche for a non-state actor like OARS to step in with less expensive yet quality-controlled methods. By using trained volunteers to collect samples, OARS estimates it saves approximately 62% in staff costs In 2020, OARS volunteers spent an estimated 225 h to be trained and conduct sampling and staff time was 83 h; without volunteers, staff time would have been an estimated 220 h [50]. Although MassDEP cited OARS data in its 2001 Water Quality Assessment Report [51], it was not until 2018 that the State began to use this data generated by an independent organization in developing its biennial Impaired Waters List. Additionally, MassDEP was allocated additional state funds in 2018 to give grants to watershed organizations to initiate or improve bacteria monitoring programs. OARS was able to take advantage of this and has completed its second year of volunteer-based *E. coli* bacteria monitoring and source tracking.

Despite the cost-effectiveness of a community science-based monitoring program, staff time and lab fees currently total over $60,000 per year. OARS requires financial support from other stakeholders. These include the National Park Service through its Wild and Scenic Rivers Program, contributions from several municipalities ($1000–2000 each), a state budget amendment annually filed by watershed legislators who see that OARS' data are filling a critical gap for decision-makers, occasional grants from community and family foundations, an environmental penalty payment, and OARS members. Only two sources have provided regular support: The National Park Service and substantial unrestricted contributions by OARS members. OARS' member support shows that OARS' science-based advocacy and the monitoring work itself is deeply embedded in the community's priorities.

## 4. Results: Progress in Restoring River Health

### 4.1. Musconetcong River

The removal of the Finesville dam set in motion the removal of the Hughesville dam 3.8 km upstream. The QAPP was revised and served as a model for assessing the Hughesville dam removal and several other restoration efforts in the Musconetcong watershed. In response to volunteer questions about the effects of restorations and dam removals

on freshwater mussels, MWA looked at the cost/benefit of streambed manipulation on these species through a three-year study funded by NRCS [52], which supported NJDEP to require mussel surveys prior to some dam removals. As dam removals became more common, MWA shared monitoring methods and protocols with other non-profits around the state.

In 2016, the Hughesville dam removal was a highly publicized event attended by federal and state dignitaries. After the removal, the site became a popular fishing spot and NJDEP started planning land acquisitions adjacent to the old industrial site. In spring the next year, anglers reported seeing small schools of American shad entering the Musconetcong. In June 2017, American shad were caught upstream of the old Hughesville dam site. It was the first time the species had been documented in the Musconetcong River in over a century and over 300 years since shad had been seen upstream of Finesville. Eric Schrading of the USFWS stated: "We must continue the critical work of the Musconetcong River Restoration Partnership to restore free-flowing waters that support water quality, wildlife, and recreational fishing and boating, and reduce the danger and costs associated with dams" [53].

In November 2017, MWA received funding to partner with the US Fish and Wildlife Service to study the impacts of the dam removals on aquatic habitat, assess the removal of the Hughesville Dam and lay the groundwork to track the progress of the largest dam removal planned on the river at Warren Glen. In 2018, the single municipality that had been reluctant to sign onto the River Management Plan partnered with MWA to apply to the National Park Service for the additional 6.9 km to be designated Wild and Scenic.

In 2020, MWA began the feasibility study for removing the Warren Glen dam. Removing this 10-m dam and the upstream Bloomsbury dam (scheduled for 2021) will complete removal of the four largest obsolete structures blocking river continuity since the river was designated Wild and Scenic. This will restore free-flowing conditions to approximately 19 km of the river and pave the way for additional restoration and durable protection of the Musconetcong River.

### 4.2. Sudbury, Assabet and Concord Rivers

Despite the continuing problem of biomass in the Assabet, there has been progress. By 2002 the data needed to analyze the problem, and identify and advocate for appropriate solutions, were in hand. OARS was able advocate for precedent-setting phosphorus limits that dramatically reduced point source contributions of nutrients based on the science. The formerly eutrophic segments of the Concord River, 50% of the flow of which is from the Assabet River, were removed from the 2016 Impaired Waters (303d) List for phosphorus, an action supported by OARS' data.

The effects of all pollutants are exacerbated by the increasing frequency and intensity of droughts and heatwaves, and intensity of precipitation events brought by climate disruption. Current predictions for extreme heat in the SuAsCo watershed estimate that days per year above 32 °C will increase from the historical level of 10 days to 40 days by midcentury, and temperatures above 38 °C will increase from zero days to 13 days by midcentury [54]. Intensity of precipitation is expected to continue its historic upward trend, with Massachusetts experiencing some of the greatest increases nationally [55]. More intense precipitation events increase stormwater runoff and pollution, reduce recharge of groundwater and baseflow, and as a result exacerbate low flow drought conditions. The watershed experienced severe to extreme drought in 2016 and again in 2020 [56,57].

In 2020, when drafting the NPDES permit for the Westborough wastewater treatment plant to discharge up to 29 million liters of effluent per day to the headwaters of the Assabet River, EPA permit writers had to use a dilution factor of zero for the first time ever due to the extremely low flows during the drought of 2016. The 7Q10 low flow metric is used to determine effluent dilution at the point of discharge into a surface water, in this case the Assabet River, so that the discharge does not cause or contribute to a water quality impairment [58]. More frequent low flows and hot weather are expected

to increase biomass production and decrease dissolved oxygen in the river resulting in increasing frequency of fish kills, toxic cyanobacteria blooms, other unpleasant conditions for recreation, and diminished scenic beauty. The water quality improvements may, for a time, help offset the worst of these climate challenges [59]. While some further point source reductions are feasible, it is clear that the focus needs to start including non-point sources and stormwater point sources.

The designation of the Sudbury, Assabet and Concord Wild and Scenic River in 1999 allowed consistent funding to flow to OARS' water quality monitoring program. It is notoriously difficult to secure grants for on-going programs like monitoring, which requires consistency rather than innovation. Consistent National Park Service funding enabled OARS to develop an uninterrupted longitudinal data set for use by regulators, municipalities and advocates. Wild and Scenic designation also gave legal standing to the National Park Service to comment on NPDES effluent discharge permits on all three rivers, covering a total of seven wastewater treatment plants. Although the worst sources of effluent were on dammed segments of the river above the designated Wild and Scenic River, up-stream pollution was flowing into the free-flowing segments downstream. Lastly, the federal designation increased OARS' ability to participate in decision-making—the fate of these rivers was a national concern and had the attention of Washington as well as of state lawmakers.

### 4.3. Co-Benefits of Community Science

The community science approach yielded multiple benefits. Figure 5 shows the participants' views of some of these benefits. OARS' team of up to 35 local volunteers became the eyes and ears of the watershed, sourcing photographs of conditions and violations that staff could follow up on, and advocating for action in person at hearings, personally to their legislators, in the Press, and through social media. Volunteer alerts from the field have identified storm drains illegally discharging sewage and streams filled with gravel to make illegal stream crossings.

A survey of MWA and OARS volunteers in October 2020 (sent to 119 with 51 respondents), showed that 85% had participated in monitoring water quality, 43% bacteria, 32% macroinvertebrates, 26% coldwater streams, 9% did data analysis; and other activities had 1–2 participants. Some participated in more than one activity.

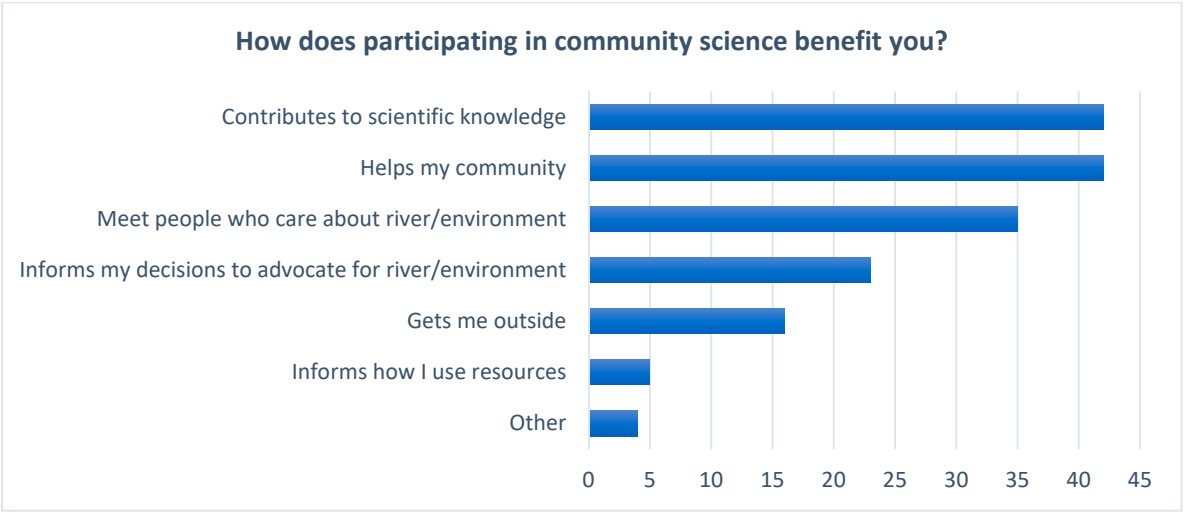

**Figure 5.** *Cont.*

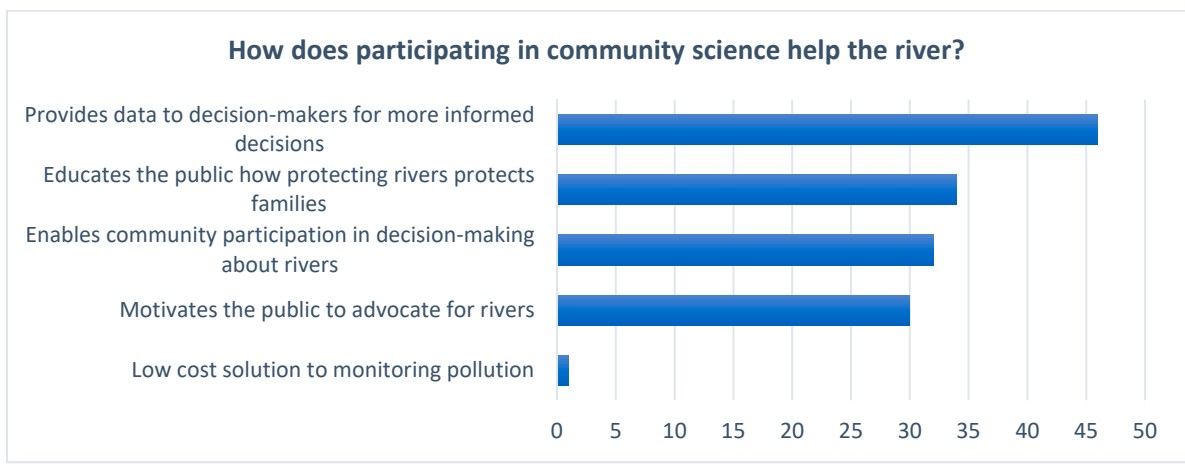

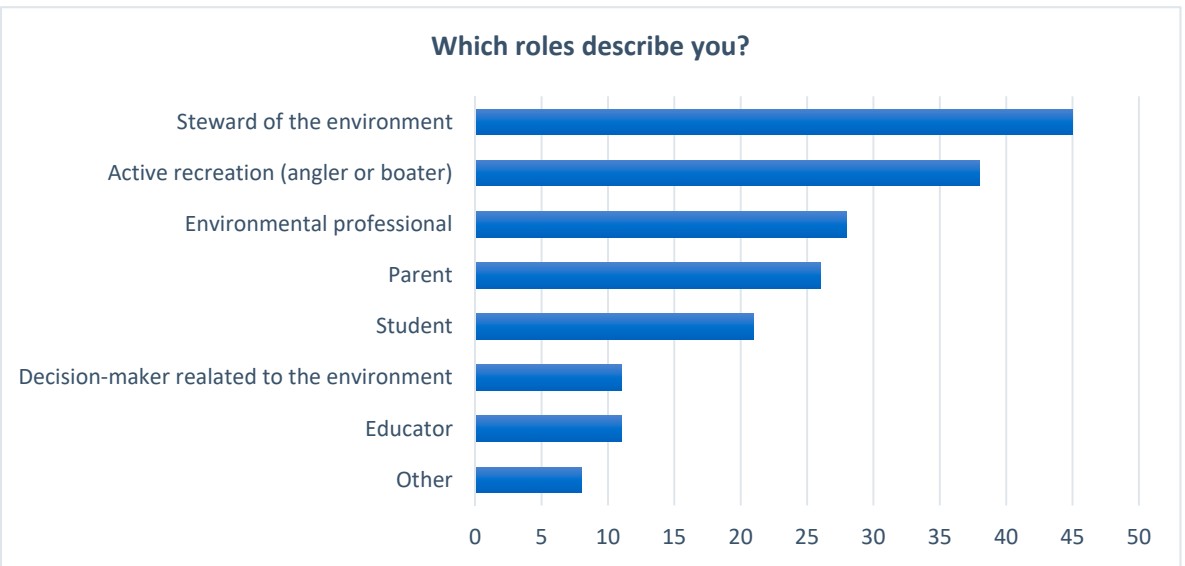

**Figure 5.** Community science survey shows percent of respondents motivated to engage in river protection. Source: OARS and MWA unpublished survey, 2020.

Forty-five percent considered themselves a "steward of the environment" and less than half actively used the river for recreation. Ages ranged from high school students to retirees (the largest occupational cohort), with 28% being environmental professionals and 11% teachers. The gender balance was about 45% female and 55% male. Many community science volunteers in these localities are very engaged in municipal decision-making—serving on local conservation commissions, planning boards, agencies' environmental commissions, or in other municipal decision-making roles—or as advocates in hearings before those boards, or in non-profit organizations. For students, participating in community science gives them work experience and contacts that improve their job and graduate school opportunities.

Currently, only one out of 44 community science volunteers at OARS and one out of 25 at MWA are people of color. The lack of diversity only partially reflects the watershed communities. In both cases, the primarily white suburban or rural watersheds also have some diverse small urban areas, and the towns and cities have more recently seen a growing proportion of people of color.

## 5. Discussion

Doyle [60] posits that river management in the US requires three characteristics: adaptive management practices based on experience; overlapping management roles between agencies which includes "intentional redundancy and interagency competition;"

and federalism (devolution of responsibilities between national, state, municipal and other government units). Rather than being inefficient, he argues, this enables responsiveness to different social needs. These case studies support that proposition and take it one step further to include non-state actors. The case studies show that government agencies and non-profit watershed organizations have had similar action-oriented goals for their data collection, but different strengths and access to resources that, together, have resulted in the needed river management. The Partnership Wild and Scenic River system has enabled small non-profit watershed organizations to be extremely effective at bringing together partners, funding, and communities to make significant progress in restoring river health and protecting their free-flowing nature. Federal designation and federal funds have made it possible for watershed organizations to advance from their grassroots origins, yet they have retained active local connections through their community science approach. By becoming part of the decision-making process, watershed organizations have been able to press for and achieve the restoration of river health and engage community members in river stewardship.

There are challenges to creating partnerships between government agencies and non-profits, due to differences in institutional structure, physical infrastructure (e.g., availability of laboratory space), and roles. In both case studies, the non-profits found that government partners may not be able to share some data or proprietary information without a long process of formal approval or legal review, and may be required to appear neutral. In addition, non-profits did not have the access to certified analytical laboratories that government agencies had, and incurred expenses that required additional funding. These issues can be resolved in most cases with good planning or additional alliances. In New Jersey, for example, some academic institutions provide government-certified laboratory services to non-profits. When each entity acknowledges strengths and limitations, and works to build relationships across sectors, problems can be resolved. Collaboration, such as to develop a QAPP and get it approved, requires building a relationship, being respectful even when adversarial, and humility about one's own knowledge and capacity. Essential elements of successful volunteer-based monitoring such as annual training of volunteers, acquiring equipment, and building staff expertise also require planning and thoughtful partnerships.

These case studies further show that this collaboration between state and non-state actors was made possible by using community science to collect consequential data. In both cases, volunteers collected critical data that guided decision-makers towards the most appropriate actions when the science wasn't settled. In the case of the MWA, the data showed the unexpectedly rapid response of certain invertebrate species to dam removal. In the case of OARS, the water quality data paired with biomass measurements showed that non-point sources needed renewed attention if "adaptive management" was to meet its goal. These outcomes were only possible, however, once transparent quality-control mechanisms like QAPPs were put in place.

In addition to gaining useful data and saving public funds, community science enabled watershed residents to become more engaged and educated [61]. Volunteers highly valued the opportunity to provide sound data to decision-makers as well as educate their community [62]. While some had a scientific background, many did not. With the current trend of the validity of science being questioned in the public realm, every opportunity to facilitate the public's understanding of the legitimacy and practical use of data is valuable.

This is particularly relevant when looking at how to ensure that the benefits of river health improvements equitably serve the diverse local populations. As discussed above, "community science" should be both by and for the community. Upstream communities may have quite different interests from downstream communities—urban vs. rural, polluter vs. polluted, wealthy vs. income-constrained. It is important to build understanding that river communities share responsibilities towards each other. In many cases, pollution control and river restoration can have major up-front costs despite the long-term benefits. Finding the most equitable way to pay for these actions—geographically,

temporally and demographically—requires some difficult conversations and confronting longstanding biases.

Watershed organizations provide a unique opportunity to bring the diverse stakeholders together in a process that develops evidence to support effective solutions at the most local level. The agency-level partnerships are usually the result of governmental planning and strategies—watershed organizations can only insist that they have a role and equal status. This paper suggests that the benefits of community science could be expanded further by engaging more diverse teams of volunteers in the research questions themselves. Including more perspectives may create a better understanding of problems and lead to more effective and equitable solutions.

Clearly, work remains to be done, as the current corps of community science volunteers in both organizations have few people of color or those with limited financial resources, although they do have a range of ages and a gender balance. This will take another level of partnership and conscious long-term efforts to discover topics of mutual concern and new ways to discover and work with diverse partners and listen to each other.

There are challenges and limitations to community science too. Volunteers need to be encouraged and adequately trained; it may not always seem an efficient way to collect data. Significant staff time needs to be devoted to identifying data quality issues. The QAPP provides a strict set of instructions, minimizing subjectivity, but volunteers have to be aware of its contents. Developing the QAPP requires expertise on the part of staff charged with the task, and would be required even if not using volunteers.

Partnership Wild and Scenic Rivers provide an opportunity to increase interaction with diverse and more urbanized communities due to their location outside of large tracts of federally-owned land. This can result in more equitable access to, and enjoyment of, these outstanding resources, providing the benefits of nature's biodiversity for mental and physical wellbeing, education and professional development. Creating a broader base of financial support can also provide more of the all-important investment in restoration of free-flowing conditions and river health.

Both the Musconetcong and the Sudbury, Assabet and Concord Wild and Scenic Rivers were designated for scenic, recreational and cultural values, not because they were "wild." For recreation to flourish and scenic beauty to endure a changing climate, efforts to reduce water pollution and improve flow and river continuity must be redoubled. Partnerships at all levels are key to achieving this. In Massachusetts, since stormwater is now considered the "leading water pollution source, causing or contributing to at least 55% of impairments in all Massachusetts' assessed waters" [63], the focus is shifting to non-point sources. For both river systems, an effort must be made to continue to remove outdated dams to restore river continuity and remove the impact of legacy nutrients in sediment.

## 6. Conclusions

Whether the community science approach can be applied beyond these two cases will depend on the local context, the regulatory framework, and the availability of potential partners. The eastern United States has a strong tradition of local home rule, with management of local governance and social services by community volunteers, which is not the model throughout the world. Nevertheless, the successes illustrated by the case studies could not have been achieved without a strong national-level framework of environmental laws: The Clean Water Act and the Wild and Scenic Rivers Act. In many countries, especially those with nascent environmental laws or restricted nongovernmental participation, advocacy for appropriate regulatory frameworks and funding will be needed to bring together the necessary elements. Government agencies in some countries may be less open to collaboration or have fewer resources to share. In countries without a strong tradition of volunteer public service, the public may expect a greater government role and be less likely to volunteer.

Both OARS and MWA continue to advocate for policy and regulatory change to improve water resource management. This is needed at the municipal, county, state or

national level, or most likely at every level, and the methods will be highly case-specific. Whatever the local conditions, the authors show the importance of establishing a set of shared goals among partners and identifying the ways in which healthier rivers benefit each set of interests.

These cases show the benefit of engaging a full range of stakeholders and considering a full range of options to improve river conditions. However, this is not enough without scientific data to provide the proof of both the causes of the problem and the likely benefits of proposed solutions. Community science fills a key niche by being both the most cost-effective and engaging approach. Rivers need long-term stewardship, yet-long-term funding and even continued institutional existence is not guaranteed. Having an informed corps of volunteers who live within the watershed helps to maintain engagement and local credibility that is needed in the long run. Even if it cannot be prepared at the outset, quality control measures that are strictly adhered to are essential. These two cases show that a local group of committed people can start with limited resources, and then build a robust high-quality and data-driven contribution to enduring river protection.

**Author Contributions:** These authors contributed equally to this work. Conceptualization, methodology, investigation, writing—original draft preparation, review and editing, A.F.-J. and N.R.-L.; data curation, A.F.-J. Both authors have read and agreed to the published version of the manuscript.

**Funding:** Research and publication of this article was funded in part by The U.S. National Park Service through the Sudbury, Assabet and Concord Wild and Scenic River Stewardship Council (SuAsCo RSC), the Commonwealth of Massachusetts through MassDEP, and contributions by OARS members, and the National Fish and Wildlife Foundation through a cooperative agreement with the United States Fish and Wildlife Service grant #55098 to the MWA, The Watershed Institute Grant Program 2009, and contributions by MWA members.

**Data Availability Statement:** OARS water quality data are available through the EPA's Water Quality Portal (WQP), available at: https://www.epa.gov/waterdata/water-quality-data-download#portal (accessed on 3 February 2021). OARS Annual water quality and biomass reports are available at: http://www.oars3rivers.org/river/waterquality/reports (accessed on 3 February 2021); further data are available at the OARS River Report Card: https://ecoreportcard.org/report-cards/sudbury-assabet-concord-rivers (accessed on 3 February 2021). MWA data are available upon request at info@musconetcong.org.

**Acknowledgments:** The authors would like to acknowledge Beth Styler-Barry, former Executive Director of the MWA, and Suzanne Flint, former Staff Scientist at OARS, for their vision and work to develop the first QAPP for the Musconetcong and SuAsCo, respectively, and to engage community scientists. The authors thank Julie LaBar, Pete Shanahan, Kim Kastens, David Pincumb, Ben Wetherill, and Suzanne Flint for their helpful comments, and Christa Reeves and Ben Wetherill for their help with graphics. OARS is grateful to Massachusetts State Senator Michael Barrett and Representative Kate Hogan for their commitment to securing funding for water quality monitoring in the state budget. The authors recognize Julie Vastine, Director of Alliance for Aquatic Resource Monitoring (ALLARM), and Erik Silldorf, Restoration Director and Senior Scientist at Delaware Riverkeeper, for their support in adapting the management of dam removal monitoring.

**Conflicts of Interest:** Nancy Roberts-Lawler declares no conflict of interest. Alison Field-Juma is the Vice Chair of the SuAsCo Wild and Scenic River Stewardship Council which provided partial funding for this research and article. The funders had no role in the design of the study; in the collection, analyses, or interpretation of data; in the writing of the manuscript, or in the decision to publish the results.

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
