# Peer review of "Using Partnerships and Community Science to Protect Wild and Scenic Rivers in the Eastern United States"

_sustainability, doi:10.3390/su13042102_

Round 1

Reviewer 1 Report

GENERAL COMMENTS

This is a very interesting well-written manuscript about the use of citizen and community science on 2 cases studies on river restoration in the US. However, there are 2 aspects that first need to be dealt before a decision:

  • Length of the paper. It seems excessive and the manuscript is too descriptive in my opinion. Authors should consider in shortening it in at least 30%. This can be done without losing important information.
  • Most importantly, the manuscript while focusing on 2 local case-studies, clearly lacks extension to a broader scope. What key messages should the readers take home with this manuscript? In what way can be important for other contexts other than yours? (in other words, make yourself the following question: How can the findings of my study be important for readers from other contexts?). For this, a brief conclusion section should be added after the Discussion. Last sentence of the abstract could also improve to reflect this extension/application to other contexts.

SPECIFIC COMMENTS

Table 1 and throughout the manuscript – W&S ? Provide full name for acronyms upon first citation

L172 – Give complete captions for tables and figures (goal of the study, study area, etc.).

L281 – The rest of the sentence seems to be missing, otherwise check punctuation.

L352-353 – Check readability of this sentence.

Figures 5 and 6 should be re-drawn in a better quality. Also, remove limits from both figures.

Figure 8 – include units of the x-axis. %?

L679 – Discussion or Conclusion? Not both. It is clearly a Discussion, but a Conclusion would be needed also, please see General Comments.

L725 – This was not shown on the Results. Please include it or remove the sentence.

Author Response

Thank you for your excellent and thoughtful comments. We found them very helpful in refining the article.  

Length of the paper. It seems excessive and the manuscript is too descriptive in my opinion. Authors should consider in shortening it in at least 30%. This can be done without losing important information.

Substantial cuts were made in the descriptive sections, particularly Section 3.1, and language tightened up throughout. Unnecessary figures (photos) were removed. Adding a conclusion and response to some other review comments added some length back. 

Most importantly, the manuscript while focusing on 2 local case-studies, clearly lacks extension to a broader scope. What key messages should the readers take home with this manuscript? In what way can be important for other contexts other than yours? (in other words, make yourself the following question: How can the findings of my study be important for readers from other contexts?). For this, a brief conclusion section should be added after the Discussion. Last sentence of the abstract could also improve to reflect this extension/application to other contexts.

A Conclusion has been added that aims to extend the applicability to other contexts globally. Since the article did not go into any depth regarding other contexts this was not included in detail in the abstract.

SPECIFIC COMMENTS

Table 1 and throughout the manuscript – W&S? Provide full name for acronyms upon first citation.

Checked throughout and corrected.

L172 – Give complete captions for tables and figures (goal of the study, study area, etc.).

MDPI instructions for Figures that states “A caption on a single line should be centered.” The revision attempts to be complete yet not exceed one line unless necessary.

L281 – The rest of the sentence seems to be missing, otherwise check punctuation.

Sentence deleted.

L352-353 – Check readability of this sentence.

Changed to: Volunteers collected habitat information including underwater photographs in the impounded areas.

Figures 5 and 6 should be re-drawn in a better quality. Also, remove limits from both figures.

New clearer images provided, limits removed.

Figure 8 – include units of the x-axis. %?

Added to caption (percent of respondents).

L679 – Discussion or Conclusion? Not both. It is clearly a Discussion, but a Conclusion would be needed also, please see General Comments.

We added a Conclusion. We retained a shortened version of the Discussion, indicated as acceptable in the overall review comment above and as suggested by the other reviewers.

L725 – This was not shown on the Results. Please include it or remove the sentence.

This has been added in the Results.

Reviewer 2 Report

The manuscript is very well written and contains fascinating stories about two organizations and where they fit into the broader scheme of water management in their respective regions.

In my opinion, it seems as though the authors are seeking to tell comprehensive histories of their organizations and regional approaches to water management. While interesting stories, I do not believe that all of the included details contribute to the main points as declared in the title and abstract. I recommend that the article be revised with substantial condensing of extraneous details that are not absolutely critical to the telling of the community science component of the story.

Alternatively, if it is the authors’ goal to tell the full story of their organizations, then I recommend that they remove mention of community science from the title and instead include it as one of the many participants of the broad networks described here. Community science is absolutely critical in general and the intent of my comment is not to downplay community science but rather to do the opposite and strip as much of the extra information as possible so that the community science story becomes more central. I was very excited to learn about the community science aspect of this work but it seems to be a small component of the article as written.

Section 3 in particular needs the most condensing. I recommend a very short intro paragraph or two (sections 3.1.1. and 3.2.1.), but not a > two page historical overview. I also recommend that this short intro be followed by getting quickly into the approach in both case studies. My suspicion is that the two coauthors semi-independently authored the two case studies as they have rather different styles. The style of Case Study 2 is more concise and easier to read though I recommend condensing this one as well.

Here are a few more specific notes:

L 16: Could you add a short note in the abstract touching on how community-based science played a role? It is a large jump to go from “using community-based science to provide actionable data” to the end results without mentioning anything about the community-based science methods in between.

L 38: the phrase “and as devolved to the state level” is a bit awkward. Reword, especially for readers who may not be experts in the CWA and state involvement therein.

L 41: need citation for return of American shad.

L 45: the rest of the first intro paragraph is about river management. I would recommend against introducing community science in this paragraph.

L48–56: citations needed

L67: is there a word missing in this section header? Doesn’t read smoothly; if “Partnership Wild and Scenic” is a specific name, I imagine it will be introduced later?

L198: end of the sentence “the shad stopped…” doesn’t fit in this sentence as written.

L204–206: citation needed

L213: “were state-classified” and no comma needed after Musconetcong

L217–218: sentence not needed and too conversational a tone in my opinion

L232–257: much of this section is anecdotal and can be substantially condensed without losing the main messages about the need for high quality data. Please condense to just a couple sentences.

Fig. 5, 6 have to be substantially higher quality

Author Response

Thank you for your excellent and thoughtful comments. We found them very helpful in refining the article.  

The manuscript is very well written and contains fascinating stories about two organizations and where they fit into the broader scheme of water management in their respective regions.

In my opinion, it seems as though the authors are seeking to tell comprehensive histories of their organizations and regional approaches to water management. While interesting stories, I do not believe that all of the included details contribute to the main points as declared in the title and abstract. I recommend that the article be revised with substantial condensing of extraneous details that are not absolutely critical to the telling of the community science component of the story.

Alternatively, if it is the authors’ goal to tell the full story of their organizations, then I recommend that they remove mention of community science from the title and instead include it as one of the many participants of the broad networks described here. Community science is absolutely critical in general and the intent of my comment is not to downplay community science but rather to do the opposite and strip as much of the extra information as possible so that the community science story becomes more central. I was very excited to learn about the community science aspect of this work but it seems to be a small component of the article as written.

The authors have modified the title to reflect the synergy between partnerships and community science (the second suggestion), as well as substantially condensing the background details.

Section 3 in particular needs the most condensing. I recommend a very short intro paragraph or two (sections 3.1.1. and 3.2.1.), but not a > two page historical overview. I also recommend that this short intro be followed by getting quickly into the approach in both case studies. My suspicion is that the two coauthors semi-independently authored the two case studies as they have rather different styles. The style of Case Study 2 is more concise and easier to read though I recommend condensing this one as well.

Substantial cuts were made in the descriptive sections, particularly Section 3.1, and language tightened up throughout.

Here are a few more specific notes:

L 16: Could you add a short note in the abstract touching on how community-based science played a role? It is a large jump to go from “using community-based science to provide actionable data” to the end results without mentioning anything about the community-based science methods in between.

Abstract revised to strengthen the community science connection.

L 38: the phrase “and as devolved to the state level” is a bit awkward. Reword, especially for readers who may not be experts in the CWA and state involvement therein.

Changed to: The CWA is implemented jointly by federal (national) and state environmental agencies: Environmental Protection Agency (EPA), and New Jersey Department of Environmental Protection (NJDEP) and Massachusetts Department of Environmental Protection (MassDEP), respectively.

L 41: need citation for return of American shad.

Added.

L 45: the rest of the first intro paragraph is about river management. I would recommend against introducing community science in this paragraph.

Moved to beginning of next para.

L48–56: citations needed

Citation added and text revised.

L67: is there a word missing in this section header? Doesn’t read smoothly; if “Partnership Wild and Scenic” is a specific name, I imagine it will be introduced later?

Changed to:

2. The Partnership Wild and Scenic Rivers framework

2.1. Using the Wild and Scenic Rivers Act for partnerships

L198: end of the sentence “the shad stopped…” doesn’t fit in this sentence as written.

Changed to: Blocked from their natural habitat, the shad stopped spawning in the Musconetcong.

L204–206: citation needed

Citation added.

L213: “were state-classified” and no comma needed after Musconetcong

Corrected.

L217–218: sentence not needed and too conversational a tone in my opinion

Changed to:

Dam owners faced liability for dam failure and/or prohibitive permit and construction costs for repair. Some sought the advice of the MWA. 

L232–257: much of this section is anecdotal and can be substantially condensed without losing the main messages about the need for high quality data. Please condense to just a couple sentences.

Condensed but retained information about the role of partnerships and their differing needs for data.

Fig. 5, 6 have to be substantially higher quality

Replaced with high quality images.

Reviewer 3 Report

This work deals with 2 cases involving non-profit watershed-based organizations in order to an approach for management challenge directed to develop durable protections for rivers. This paper starts with an analysis of the Partnership Wild and Scenic rivers program where is treated the importance of local watchdog groups in monitoring water quality for the purpose of river protection but the requirement that the data obtained follows the framework of Quality Assurance Project Plans (QAPP). The mentioned case studies are distinct: one case deals with the ecological impact of dam removal and the other one the control of water quality in a stream impacted by effluents from wastewater treatment plants originating an excessive biomass.
The paper is interesting, with a convenient structure, even if sometimes seems too detailed, but, on contrary, I find that the discussion is rather incomplete: What are the limitations of using community science? What were the difficulties in putting together government agencies and non-profit organizations? How to overcome financing problems? And above all of them: what are the lessons obtained from the case studies and how to extend these actions to another areas?

Table 2 It is required a more detailed information about the methods used in habitat assessment/ habitat score concerning case 1.
Line 352 Rephrase
Line 599 Why the removal of 4 obstacles, when only Finesville and Hughesville dams were considered?

Author Response

Thank you for your excellent and thoughtful comments. We found them very helpful in refining the article.  

The paper is interesting, with a convenient structure, even if sometimes seems too detailed, but, on contrary, I find that the discussion is rather incomplete:

What are the limitations of using community science?

A discussion has been added:

There are pitfalls to community science too. Volunteers need to be encouraged, trained and at times it may seem that it is not an efficient way to collect data. Quality control is essential, and significant staff time is spent poring over the data to identify any problems. However, the QAPP provides a strict set of instructions, which minimizes subjective interventions. Developing the QAPP is a major task and requires expertise on the part of the staff member and supervisor who are charged with the task but would be required even if not using volunteers.

What were the difficulties in putting together government agencies and non-profit organizations?

Revised to add discussion of this: L617-629 (p. 18)

How to overcome financing problems?

A brief discussion of the need to have many and diverse sources.

And above all of them: what are the lessons obtained from the case studies and how to extend these actions to another areas?

Please see new Conclusion.

Table 2 It is required a more detailed information about the methods used in habitat assessment/ habitat score concerning case 1.

Method details have been added.

Line 352 Rephrase 

Changed to: A team with a boat and Ekman dredge collected, examined and preserved samples at Site 2.

Line 599 Why the removal of 4 obstacles, when only Finesville and Hughesville dams were considered?

See new Line 221-224: The Partnership began applying for federal grants targeted to remove the four larger dams in the lower Musconetcong that impeded fish migration. The Finesville, Hughesville, Warren Glen would be coordinated by MWA, and the Bloomsbury dam removal coordinated by the United States Army Corps of Engineers. 

Round 2

Reviewer 1 Report

I thank the authors for their corrections and amendments on the revised version. I am satisfied with the changes made by the authors in response to my previous concerns, and believe that the manuscript can now be accepted on its present form.

Author Response

Thank you so much--your comments have helped very much and we are pleased that the article has been accepted!

Reviewer 2 Report

This revision is well written. Many of my prior concerns have been sufficiently addressed. I still believe that the article is longer than it needs to be for the amount of original content (i.e., some parts are still a little too wordy for me) but I don't have any specific suggestions and don't think this recommendation merits holding up the publication further.

I do have a few important suggested changes, though, that should definitely be addressed prior to publication:

  • Page numbers are included in certain citations in the main text body and should be removed (multiple occurrences).
  • L394: should be "orthophosphate" or "ortho-phosphate" not "ortho-phosphorus"
  • Figs 3 and 4 have apparent linear regressions that are not described in the legend nor figure captions. These should be removed. If they are absolutely critical in your opinion—and I firmly believe they are not—they must be described in legends and figure captions and their parameters must be shown. Otherwise, I believe it is inappropriate and potentially misleading to include them. Again, I would prefer for them to be removed. 

Thanks and nice article. I learned a lot!

Author Response

Thank you for your suggestions, they have been very helpful. We also made additional cuts to the text to refine it. 

  • We moved the page numbers in the text to the references.
  • L394: We corrected it to read "orthophosphate"
  • We removed the regression lines from Figs 3 and 4.